# Response of the Coastal Phytoplankton Community to the Runoff from Small Rivers in the Northeastern Black Sea

**Valentina M. Sergeeva** [1,*] , **Sergey A. Mosharov** [1,2] , **Natalia A. Shulga** [1] , **Viacheslav V. Kremenetskiy** [1] , **Pavel V. Khlebopashev** [1] **and Dmitry N. Matorin** [3]

1   Shirshov Institute of Oceanology, Russian Academy of Sciences, 117997 Moscow, Russia
2   Faculty of Power Engineering, Bauman Moscow State Technical University, 105005 Moscow, Russia
3   Department of Biology, Moscow State University, 119234 Moscow, Russia
*   Correspondence: vsergeeva@gmail.com

**Abstract:** River runoff is an important source of nutrients as well as suspended and dissolved organic matter that in coastal zones and on the shelf are transformed due to local production cycles. River runoff affects the hydrological regime, salinity, temperature, and irradiance in river–seawater mixing zone. Our study focuses on the response of phytoplankton to the impact of small Caucasian rivers in the Northeastern (NE) Black Sea, as one of the most sensitive components of marine ecosystems with respect to the changes in abiotic factors. The leading role of marine species of diatoms, dinoflagellates, and coccolithophores in the structure and functioning when impacted by runoff from small rivers is demonstrated in comparison to the freshwater community. Variability of the taxonomic composition and quantitative and productive characteristics of marine phytoplankton communities impacted by small rivers were comparable to or exceed the seasonal and interannual variability on the NE Black Sea shelf. This indicates the significant role of runoff from small Caucasian rivers in maintaining of a high production level of phytoplankton overall and of the coccolithophore *Emiliania huxleyi* in particular in the coastal zone.

**Keywords:** marine coastal phytoplankton; community structure; small river impact; salinity gradient; NE Black Sea; primary production

## 1. Introduction

The uniqueness of the Black Sea ecosystem is governed by specific hydrological conditions. Among which are the significant continental freshwater runoff that exceeds the evaporation level, the relative isolation of the water basin, limited connection with the more saline and denser Mediterranean Sea waters, and the bottom topography—all of which prevent vertical mixing of water [1]. The average salinity in Black Sea surface water is about 18 [2].

The coastal zone of the Black Sea is the most desalinated due to significant river runoff with an average annual discharge of 348 km$^3$ per year [3]. Salinity can decrease to less than 9 near estuaries [2]. The northwestern (NW) Black Sea is the most freshened. The large Danube and Dnieper rivers contribute to about 73% of the Black Sea total annual river runoff [3]. Due to the shallow depth and remoteness from the deep basin in the NW part, river runoff forms significant freshened areas up to several tens of thousands of km$^2$ [4]. In the mid-20th century, intensification of human economic activity caused an increase in the Danube and Dnieper's nutrient supply [5]. Eutrophication led to frequent phytoplankton blooms of up to over 90 million cells L$^{-1}$ and 30,000 mg wet weight m$^{-3}$ [6,7] as well as changes in the biooptical and oxygen conditions on the NW Black Sea shelf [8,9]. In addition, harmful algal blooms have been frequently recorded here [10–13]. Paleoecological studies have confirmed an increase in species diversity and abundance of dinoflagellate cysts in NW Black Sea shelf sediments in the second half of the 20th century [14]. The

dominant cysts among dinoflagellates were *Lingulodinium polyedrum*, *Polykrikos schwartzii*, *Spiniferites* spp.—potentially toxic species capable of producing yessotoxin [15–17].

Numerous, relatively small montane rivers and streams flow into the Northeastern (NE) Black Sea. The average annual discharge here is only about 2% of the total annual river runoff into the Black Sea [3]. The fresh water interacts with the narrow shelf zone, mostly less than 11 km wide and no more than 40 km at its widest. The narrow shelf stretches along the coastline for only about 3000 km$^2$. Interaction between fresh and seawaters forms local desalinated areas called plumes with a salinity up to 16 on the surface [18]. Despite the small annual river discharge in the NE sector, there is a significant seasonal and synoptic variability in the area and distribution of plumes on the surface of the Black Sea [19,20]. Seasonal variability is associated with the type of power supply of NE rivers, i.e., rain and seasonal snow. During active snowmelt and heavy rains (mainly in spring—summer), the plumes of small rivers have a large length and areas up to 50 km$^2$ [18,21].

Joint hydrochemical and hydrophysical studies conducted annually by the Shirshov Institute of Oceanology of the Russian Academy of Sciences (IO RAS) have shown that high concentrations of nutrients and suspended particulate and dissolved organic matter enter the NE shelf with river runoff [18,22–24]. The nutrient content in the NE Caucasian rivers can reach values of 45 µM for nitrates, 4 µM for phosphates, and 130 µM for silicates [18,24]. Coastal eutrophication should impact the shallow water ecosystem and primary production conditions.

Results of long-term studies of the phytoplankton taxonomic composition in different parts of NE Black Sea, the spatial distribution of the phytoplankton abundance and biomass, seasonal dynamics, and influence of anthropogenic pressures and climate change are summarized and presented in reviews [1,25–28]. Recent studies covered phytoplankton of the NE shelf and adjacent deepwater basin of the Black Sea [29–35]. These studies have analyzed seasonal and interannual variability of the hydrophysical, hydrochemical, and biological parameters and have identified long-term trends in the relationship between abiotic factors and the variability of phytoplankton communities.

Direct phytoplankton studies in the nearshore Caucasian part of the Black Sea are numerous [36–42], investigating the species composition, seasonal variability of the abundance and biomass, the presence of potentially toxic species in communities, the appearance of new species in the community, and the impact of port conditions on plankton phytocenoses. However, the impact of abiotic factors (changes in temperature, salinity or nutrient supply) on the phytoplankton community remains unstudied. In freshwater–seawater mixing zones, there are significant variations in the hydrochemical and hydrophysical conditions. The pronounced impact of large river plumes on interannual changes in the shelf phytoplankton community of the NW Black Sea have been studied in the course of long-term projects as mentioned above. It remains questionable whether phytoplankton of the narrow NE Black Sea shelf experiences a similar impact from numerous small rivers.

The objective of our study is to determine the impact of small rivers runoff on the structure and functioning of coastal marine phytoplankton communities and to assess its force during the first half of the growing season in the Caucasian coastal zone of the NE Black Sea.

## 2. Materials and Methods

### 2.1. Research Area

The paper focuses on biological processes in the phytoplankton community in the coastal zone of the Black Sea from April to June under low river runoff conditions in order to avoid the effect of joint influence of neighboring streams.

We investigated plumes from five rivers on the Caucasian coast between Tuapse and Sochi and near the Skurjinskii Reserve: the Tuapse, Ashe, Psezuapse, Shahe, and Kodor (Figure 1). Sampling was carried out from 29 May to 2 June 2018 and 2 April 2019 (Table 1). The average water discharge of these rivers varies between 13–30 m$^3$ s$^{-1}$ (and for the Kodor, 132 m$^3$ s$^{-1}$) [3]. Generally, maximum water discharge occurs in winter and early spring

during heavy rains and snowmelt. In summer, river discharge is usually insignificant, reaching only 7–17% of the annual water discharge. For the Kodor, often in May–June and occasionally in early April, there is no high water, so we observed the background conditions with minimal river runoff impact on the coastal ecosystem.

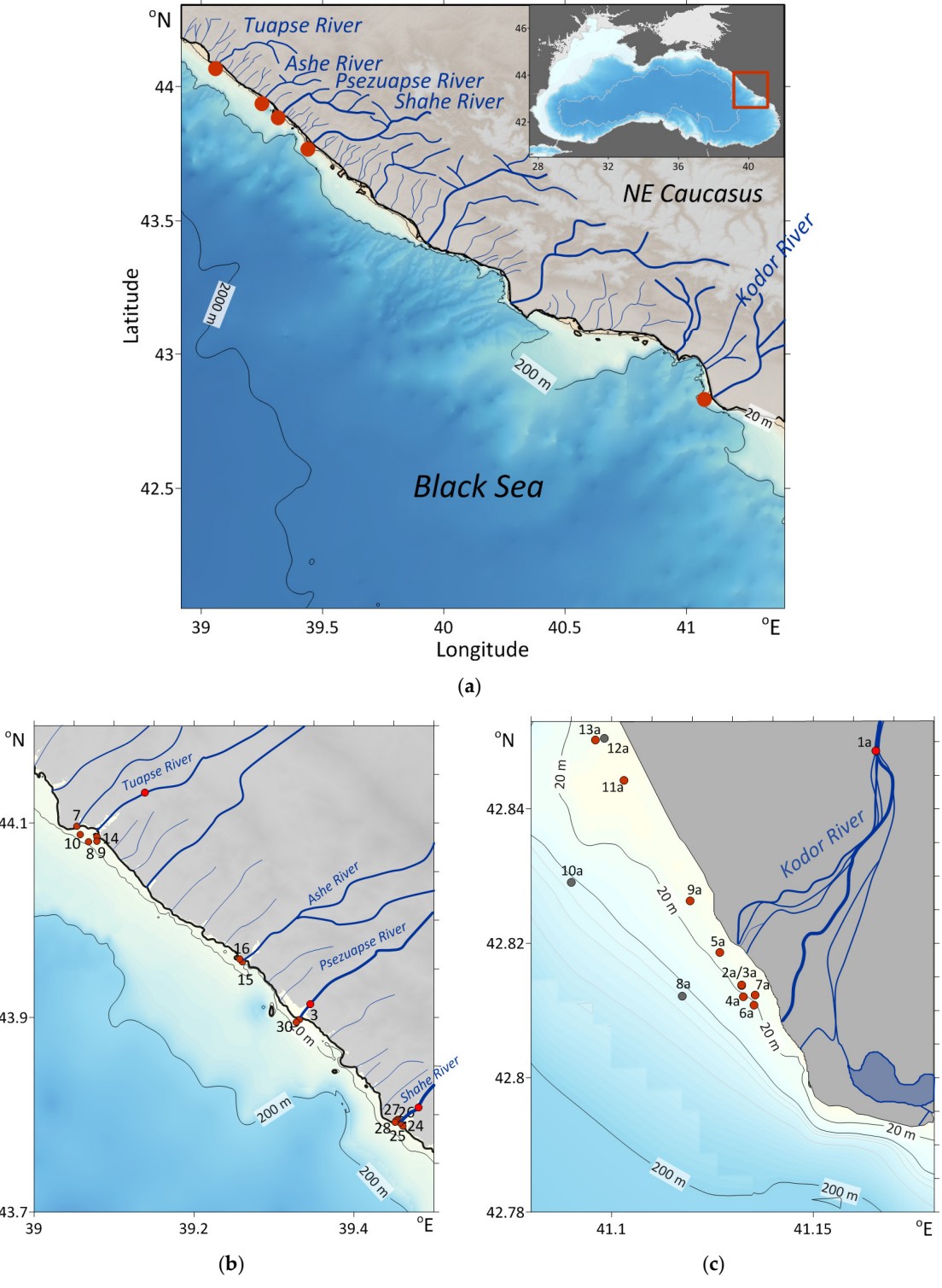

**Figure 1.** (**a**) Study area in NE Black Sea (in red frame) and positions of stations impacted by small Caucasian rivers in different seasons: (**b**) Between 31 May–3 June 2018 impacted by the Tuapse, Ashe, Psesuapse, Shahe rivers; (**c**) On 2 April 2019, impacted by the Kodor River (red points); stations 8a, 10a, 12a were outside the main stream (gray points).

**Table 1.** Study area and hydrological features during sampling (in the column "Number of Stations", the number of samples is given in parentheses taking into account sampling from the upper and bottom layers). T, temperature; S, salinity; "+", is noted in which rivers water sampling was carried out.

| Impacted by | Date | Number of Stations | Surface T Range, °C | Surface S Range | Depth at Station Range, m | Distance from Mouth, km | Sampling from River |
|---|---|---|---|---|---|---|---|
| Tuapse | 31.05.2018 | 6 (6) | 22.2–22.6 | 16.2–18.1 | 4–14 | 2.4 | + |
| Ashe | 31.05.2018 | 2 (2) | 22.7–22.9 | 16.4–17.3 | 3–4 | 0.5 | |
| Psezuapse | 29.05.2018 | 2 (2) | 22.5–23.1 | 15.2–17.7 | 3–8 | 0.6 | + |
| Shahe | 02.06.2018 | 5 (9) | 20.9–22.5 | 11.9–17.7 | 2–9 | 1 | + |
| Kodor | 02.04.2019 | 12 | 9.5–10.2 | 8.9–17.8 | 3–55 | 4.5 | + |

## 2.2. Sampling

Sampling was done from the surface layer with precise reference to the sea surface salinity, which during sampling differed in the zones of influence of small rivers (Table 1). In the zone of influence of the Shahe River, additional samples were collected from the bottom layer from depths of 3–8 m. Freshwater from each river was sampled with the exception of the Ashe River. In total, the material included 22 samples from 18 stations in May–June 2018 and 13 samples from the upper layer in April 2019.

Temperature and salinity at stations were recorded with a CTD probe (CTD—conductivity, temperature, depth, Sea-Bird Electronics 19plus) manually lowered to the bottom. In May–June 2018, sampling included biological parameters (phytoplankton and chlorophyll a concentrations and primary production rates) and nutrient content. In April 2019, in addition to these parameters, nine samples were collected for particulate organic matter (POM) and dissolved organic carbon (DOC) analysis.

For the biological parameters, water samples were collected into 1.5 L dark plastic bottles and placed on board a ship in a container filled with sea surface water.

For the hydrochemical parameters, samples were collected directly into plastic containers and immediately moved to storage containers conforming to the specific analyses [43]. Measurements were conducted within 24 h after sampling in stationary laboratory.

Surface water POM was collected in polycarbonate flasks and stored. Prior to filling, the flasks were rinsed three times with sample water. In laboratory conditions, seawater was filtered within 12 h. The water was vacuum filtered (0.25 atm) on pre-combusted (12 h, 450 °C) fiberglass filters (GF/F 0.7 μm, Whatman, Wilmington, NC, USA) 47 mm in diameter. Surface water samples for dissolved organic carbon (DOC) analysis were collected after filtration, placed into 22 mL glass flasks, acidified to pH 2, and kept at +4 °C. The filters after filtration (water volumes of 2.0 to 5.0 L) were immediately transferred with sterile tweezers to 20 mL vials with dichloromethane:methanol (DCM:MeOH 9:1 *v/v*) and kept at +4 °C prior to analysis.

## 2.3. Phytoplankton

To determine the taxonomic composition, abundance, and biomass of phytoplankton, 200 mL water samples were left to settle within 12 h at a temperature of 10 °C and were decanted 15–25 times using a 5 μm nylon mesh. The species composition, abundance, and biomass of actively photosynthesizing forms of phytoplankton were determined using epifluorescent microscope at ×100–×400 magnifications (MICMED, Saint-Petersburg, Russia) at a coastal laboratory before fixation. Analysis of phytoplankton cells in UV light makes it possible to diagnose the physiological state of algae. [44]. The technique involves determining the color and brightness of the luminescence of autotrophs. Living, actively photosynthesizing phytoplankton cells have a bright red fluorescence, and autolyzing cells have a pink or orange glow. This method makes it possible to estimate the distribution of not only total phytoplankton community along the salinity gradient but distribution of photosynthetically active cells as well.

After processing under an epifluorescent microscope samples were fixed with neutral formalin to a final concentration of 1–1.5% and processed in detail at IO RAS (Moscow, Russia). Before analysis, the samples were concentrated to 1–2.5 mL, then completely examined under a Leica DMI 4000B light microscope (Leica, Wetzlar, Germany) using Nageotte (0.091 mL) and Naumann (1 mL) counting chambers. The total species composition, abundance, and biomass, including autotrophic and heterotrophic phytoplankton, were estimated. Cell counts and taxonomic identification of algae were carried out using standard methods [45]. The cell volume was calculated based on the similarity of the cells to geometric shapes [46,47] using the measured linear dimensions. Allometric equations were used to convert wet biomass to carbon [48]. The names of algae species as well as their salinity status (marine or freshwater) have been clarified in accordance with World Register of Marine Species (WoRMS) [49].

The similarity of the taxonomic composition was assessed using the Sørensen–Dice coefficient [50] according to the formula:

$$Ks = 2C/(A + B) \times 100,$$

where C is the number of species common for the two compared groups; A is the number of species in the first group; B is the number of species in the second group.

Data analysis, verification of the relationship between the abundance and biomass of various algae groups with total quantitative characteristics of phytoplankton, salinity, and DOC as well as calculation of determination coefficients were performed using linear regression and second-order polynomial regression using Microsoft Excel. We also used statistical graphs of Grapher 11 (Golden Software, Golden, CO, USA) to analyze the vertical distribution of algae abundance and the characteristics of functioning in the surface and bottom layers in the zone of influence of the Shahe River.

### 2.4. Primary Production (PP) and Chlorophyll a (Chl a)

*PP* value of phytoplankton was determined experimentally using the radiocarbon method [51]. An $NaH^{14}CO_3$ solution was added to 50 mL flasks with water samples, followed by exposure. The light and temperature conditions were simulated in a laboratory incubator with LED illumination individually adjustable for each vial and a HAILEA-100 laboratory cooler (Guangzhou, China). Exposure lasted 3 h. After incubation, flasks were filtered onto a 0.45 μm membrane filter (Vladipore, Vladimir, Russia), dried, and delivered to the IO RAS (Moscow, Russia) to determine the radioactivity of the filters with a Triathler (Hidex, Turku, Finland) liquid scintillation counter.

The chl *a* and pheophytin concentrations were determined using the fluorescent extract method, which is currently the standard and most commonly used method [52]. Water samples in an amount of 0.5 L were passed through GF/F Whatman filters in a vacuum of less than 0.3 atm. Then, the filters were placed in 90% acetone and kept at a +4 °C in the dark for 24 h for pigment extraction. The extract fluorescence was determined before and after acidification with 1N HCl using a MEGA-25 PAM-fluorometer (Moscow State University (MSU), Moscow, Russia) [53].

The maximum quantum efficiency of Photosystem II (PSII) as an indicator of the potential photosynthetic capacity of phytoplankton was measured in water samples, kept in the dark for 20 min using also a MEGA-25 PAM-fluorimeter (MSU, Moscow, Russia). From measurements of the minimum ($F_0$) and maximum (Fm) level of fluorescence, the maximum quantum efficiency of photosystem II (PSII, Fv/Fm) was calculated as [54]:

$$Fv/Fm = (Fm - F_0)/Fm.$$

The maximum Fv/Fm value of phytoplankton under optimal environmental conditions reaches 0.70, and when photosynthesis of algae is inhibited, it decreases to less than 0.4 [55].

*2.5. Analysis of Nutrients, DOC, and n-Alkanes*

The dissolved inorganic phosphorus (P-PO$_4$), dissolved inorganic silicate (SiO$_3^{2-}$), nitrite nitrogen (N-NO$_2$), nitrate nitrogen (N-NO$_3$), and ammonium nitrogen (N-NH$_4$) concentrations were measured using standard procedures [43].

In the laboratory, POM samples were extracted using ultrasonication three times for 5 min. Extracts (total lipid extracts) were concentrated, cleaned with activated copper, and separated into hydrocarbon and polar fractions through column chromatography over SiO$_2$. In this study, we only present the hydrocarbons (n-alkanes) analysis results. GC–MS analyses of n-alkanes were performed using a Shimadzu QP5050 device (Tokyo, Japan) with a Rxi-5Sil MS 30 m × 0.25 mm × 0.25 μm capillary column. The temperature program was as follows: starting with 3 min at 60 °C, then heating to 300 °C at 4 °C min$^{-1}$, followed by holding for 30 min at 300 °C. The injection volume was 2 μL, split less. The carrier gas was helium with a flow rate of 1.5 mL min$^{-1}$. The analysis was performed as a total scan from m/z 50 to 650 (70 eV). n-Alkanes were identified and quantified using mass spectra and retention times of the calibration mixture (n-C$_{8-20}$, n-C$_{21-40}$ mixtures, Fluka, Seltze, Germany). The response factors were determined with respect to squalane (2,6,10,15,19,23-hexamethyltetracosane) as the internal standard. Concentrations of individual hydrocarbons were reported in μg L$^{-1}$ of the filtered water. In this paper, we describe the total lipid extract as organic matter. DOC was analyzed by high-temperature catalytic oxidation (Shimadzu TOC-VCPH, Tokyo, Japan). The range of the measurable DOC concentrations was 0.05–25,000 mgC L$^{-1}$. All calibrations and data analysis are described in more detail in [56].

**3. Results**

In April, the distance of the area impacted by the Kodor River was about 4.5 km long, and the data on structural and functional changes in coastal phytoplankton communities were obtained in a wide salinity range of 8.9–17.8 (Table 1). In May–June 2018, due to the insignificant length of the river plumes (less than 2.5 km), it was not possible to collect samples in all salinity gradations for all rivers. The studied salinity ranges differed (Table 1).

According to the low chl *a* content, the low *PP* rates and low levels of the maximum quantum yield of PSII (Table 2) for all rivers, except for the Tuapse, correspond to the oligotrophic type [57], which is characteristic of the majority of montane rivers. The lower reaches of the Tuapse River flow through an urban area. The phosphorus content in the river is four times higher, chl *a* is ten times greater, and the *PP* rate is 1000 times higher than for the other investigated rivers. Such indicators correspond to the mesotrophic level of freshwater reservoirs [57].

**Table 2.** Trophic conditions and nutrient content of Tuapse, Psezuapse, Shahe, and Kodor rivers.

| River | Data | T,°C | S | chl *a*, μg L$^{-1}$ | % Phaeo | PP, μgC L$^{-1}$ d$^{-1}$ | Fv/Fm | P-PO$_4$, μM | Si, μM | N-NO$_3$, μM | Trophic Level [2] |
|---|---|---|---|---|---|---|---|---|---|---|---|
| Tuapse | 31.05.2018 | 24 | 0.1 | 3.64 | 16% | 1809.5 | 0.564 | 4.4 | 98.9 | 13.2 | Meso |
| Psezuapse | 29.05.2018 | 22 | 0.1 | 0.2 | 61% | 1.6 | 0.277 | 0.6 | 107.5 | 18.6 | Oligo |
| Shahe | 02.06.2018 | 22 | 0.1 | 0.3 | 56% | 4.7 | 0.510 | 0.6 | 121.7 | 14.8 | Oligo |
| Kodor | 02.04.2019 | 9 | 0.01 | 0.1 | 82% | 0.8 | 0.284 | 1.4 [1] | 128.5 [1] | 16.3 [1] | Oligo |

[1] In related publication [24]. [2] According to [57].

Subsequently, the spatial and seasonal variability of the structural and functional characteristics of phytoplankton was analyzed with due regard to the characteristics of small rivers that form plumes in the Black Sea coastal zone. For reasons related to the fact that, in May-June, the river mouths were located at a short distance from each other (from the Ashe River to the Shahe River is about 25 km), the samples were collected in a very short time (3 days); data for the Ashe, Psezuapse, and Shahe rivers were combined. Thus, the studied salinity range for the end of May to the beginning of June was 11.9–17.7, becoming comparable to the salinity range in April (8.9–17.8). Data on the coastal plume of the Tuapse River are excluded from the general analysis and are presented separately.

### 3.1. Phytoplankton

3.1.1. Taxonomic Structure

The taxonomic structure of phytoplankton in April and May–June in the areas of influence of rivers was carried out for the same salinity ranges: 10.8–17.7 for April and 11.9–17.8 for May–June. In April, the stations with surface salinity less than 10.8 were excluded from this analysis. The species richness of coastal phytoplankton in the areas of influence of oligotrophic rivers was similar (Figure 2). In early April, we found 90 taxonomic units (species and genera) in zone of influence of the Kodor and in late May to early June, 78 taxonomic units in the zone of influence of the Ashe, Psezuapse, and Shahe rivers.

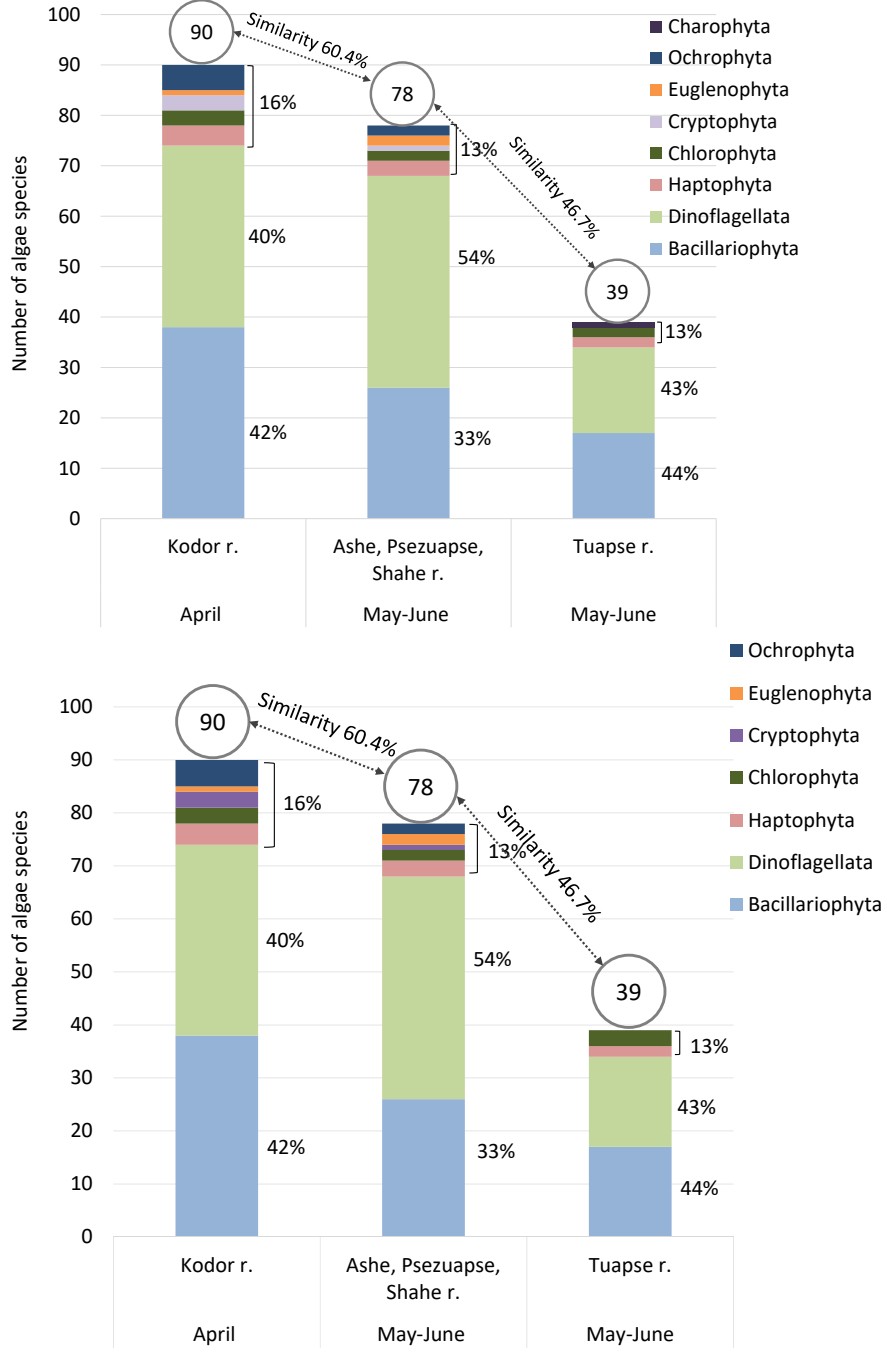

**Figure 2.** Taxonomic composition of marine coastal phytoplankton in April and May–June under the influence of oligotrophic and mesotrophic rivers of NE Black Sea. The similarity of the taxa composition is presented in accordance with the Sørensen–Dice index.

According to [58], all identified eukaryotic species belonged to eight ecological groups: Bacillariophyta, Dinoflagellata, Ochrophyta, Haptophyta, Cryptophyta, Chlorophyta, Euglenophyta, Charophyta. The similarity of the species composition according to the Sørensen–Dice index was 60%. In the plume of the mesotrophic Tuapse River, a significantly smaller number of taxonomic units were found, i.e., 39 units from five taxa that are Bacillariophyta, Dinoflagellata, Haptophyta, Chlorophyta, Charophyta. The level of similarity with the zones impacted by oligotrophic rivers was 47%.

The decrease in the number of taxonomic units could be associated with a smaller salinity range in the zone of influence of the Tuapse River. We only covered the range from 16.2 to 18.1; the inner freshened part of the plume was beyond the scope of the study.

### 3.1.2. Freshwater Phytoplankton

In May–June, despite the proximity of the stations to the mouths of oligotrophic rivers, freshwater species of planktonic algae were not found in the salinity range 11.9–17.7. In April, in the Kodor River plume, the share of freshwater species in the total abundance of phytoplankton sharply decreased from 100 to 20% with an increase in salinity from 0 to 8.9 (Figure 3). With an increase of salinity greater than 14, only empty frustules of freshwater diatoms were found; their share in the total population did not exceed 3%.

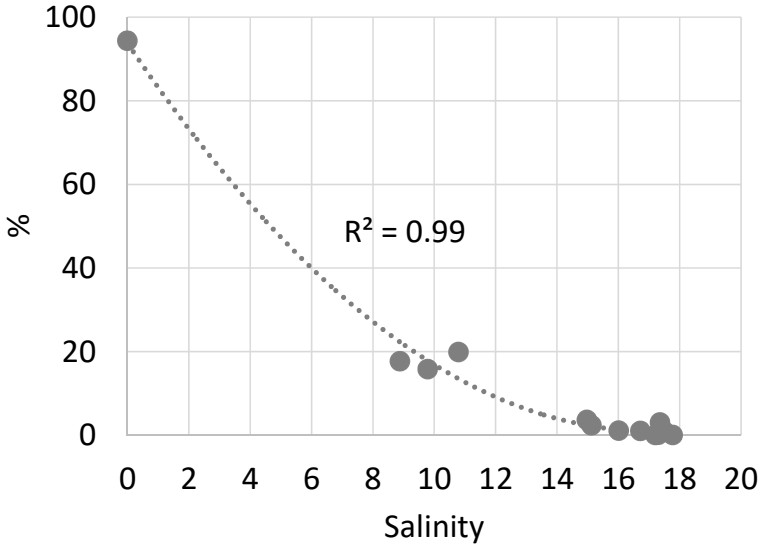

**Figure 3.** Share of freshwater species in total phytoplankton abundance (%) along salinity gradient of surface layer in April 2019 (zone of influence of Kodor river).

In the plume of the mesotrophic Tuapse River at a salinity of 16, a small number of freshwater algae cells (up to $1 \times 10^3$ cells $L^{-1}$) were found: *Cosmarium formosulum* (Charophyta) and *Desmodesmus armatus* (Chlorophyta).

### 3.2. Influence of Oligotrophic Rivers

#### 3.2.1. Total Abundance of Cells and Carbon Biomass

The range of variability of the total abundance and biomass of phytoplankton on the surface along the salinity gradient under freshwater–seawater mixing was significant. In April (influence of the Kodor River), the range of abundance was $74 \times 10^3$–$580 \times 10^3$ cells $L^{-1}$ (mean value of $207.8 \times 10^3$ cells $L^{-1}$, SD $142.4 \times 10^3$ cells $L^{-1}$). The biomass range was 14.4–31 µgC $L^{-1}$ (20.6 µgC $L^{-1}$, SD 5.3 µgC $L^{-1}$) or in wet weight 118.3–463.8 µg $L^{-1}$. In May–June (influence of the Ashe, Pszuapse, and Shahe rivers), it was $452 \times 10^3$–$974 \times 10^3$ cells $L^{-1}$ (mean value of $654.7 \times 10^3$ cells $L^{-1}$, SD $148 \times 10^3$ cells $L^{-1}$) and 40.8–67.5 µgC $L^{-1}$ (mean value of 57.3 µgC $L^{-1}$, SD 12.22 µgC $L^{-1}$) or in wet weight units 350.6–817.7 µg $L^{-1}$ (Figure 4).

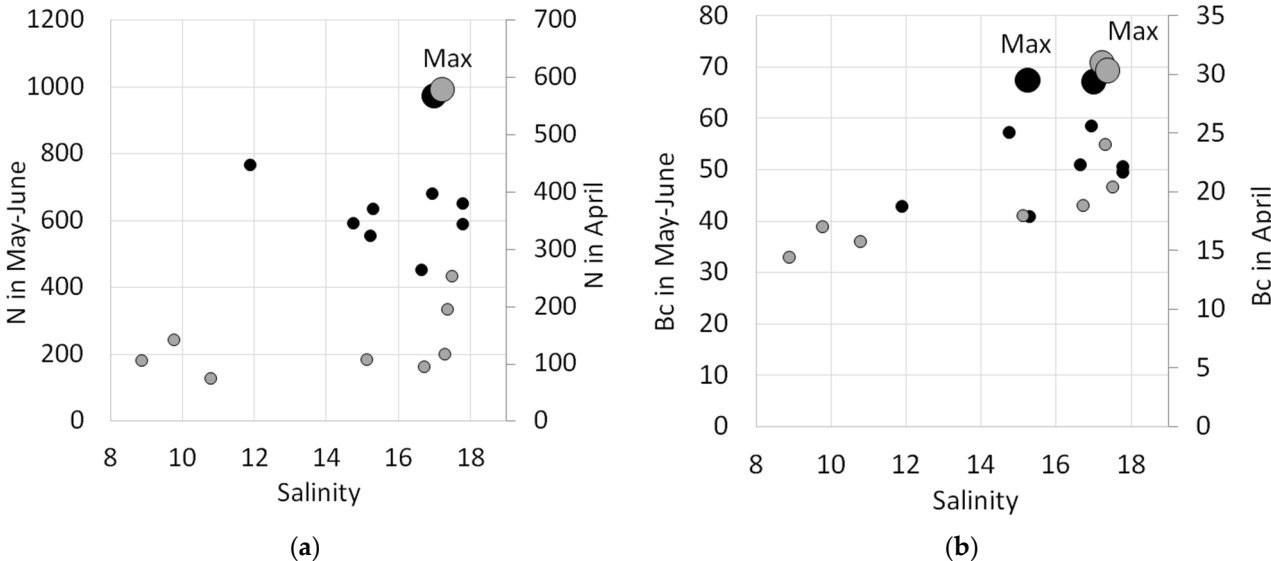

**(a)**　　　　　**(b)**

**Figure 4.** Variability of quantitative characteristics of phytoplankton along the salinity gradient under river–seawater mixing: (**a**) total abundance (N, $\times 10^3$ cells L$^{-1}$) and (**b**) total carbon biomass (Bc, µgC L$^{-1}$). ⬤—in April (under Kodor influence), ⬤—in May–June (under influence of Ashe, Psezuapse, and Shahe rivers).

The maximum peaks of the total abundance and biomass in April and May–June were observed at salinity 17–17.2. In May–June, the second maximum of the phytoplankton biomass was observed at a salinity of 15.2. In the marine zone, with an increase in salinity greater than 17.8, the abundance and biomass decreased by about 1.5–3 times.

Changes in the total abundance of phytoplankton were largely determined by coccolithophores, and the total biomass was determined by dinoflagellates with a rather high determination coefficient (Figure 5).

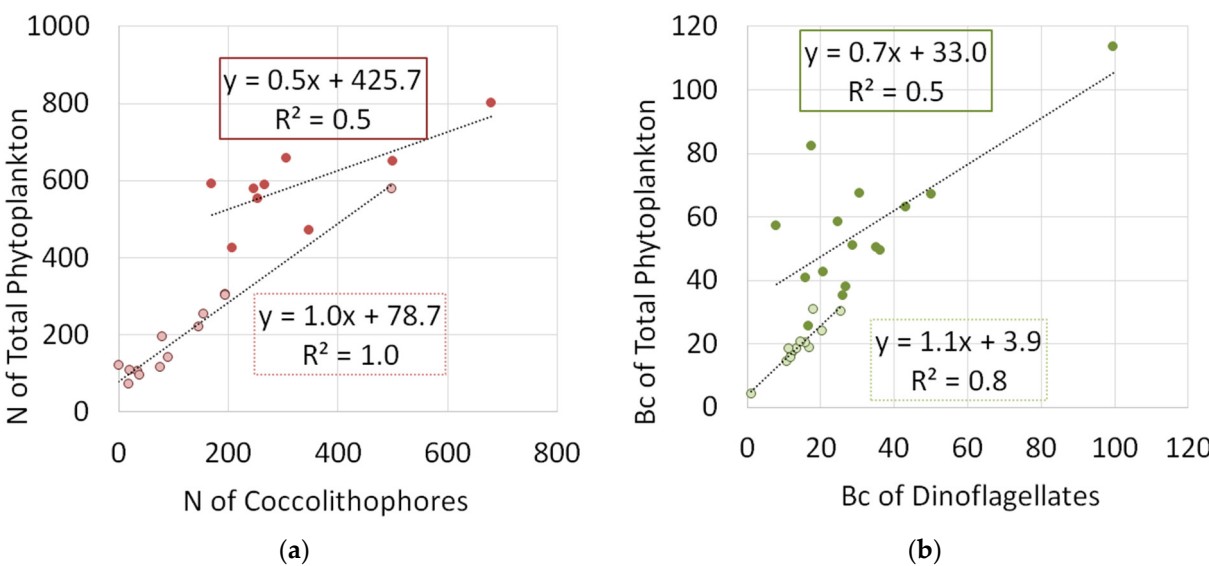

**(a)**　　　　　**(b)**

**Figure 5.** Relationship between total quantitative characteristics of phytoplankton with (**a**) abundance of coccolithophores and (**b**) biomass of dinoflagellates in zones of influence of small rivers. Pale circles, for the influence of the Kodor River (in April); dark circles, for the influence of the Ashe, Psezuapse, and Shahe rivers (in May–June). N, abundance, $\times 10^3$ cells L$^{-1}$; Bc, carbon biomass, µgC L$^{-1}$.

### 3.2.2. Spatial Distribution of Diatoms, Dinoflagellates, and Coccolithophores

Here, we consider the distribution of the main ecological groups of phytoplankton: diatoms, dinoflagellates, and coccolithophores. Their total contribution to the abundance and biomass at all studied coastal stations exceeded 70 and 85%, respectively. The distribution of the abundance and biomass of diatoms, dinoflagellates, and coccolithophores along salinity gradient in the zones of influence of oligotrophic rivers has not always been ordered. In April, a low determination coefficient indicated the absence of a strong relationship with salinity (Figure 6a,b). In May–June, there was a more distinct relationship between the different groups and salinity (Figure 6c,d).

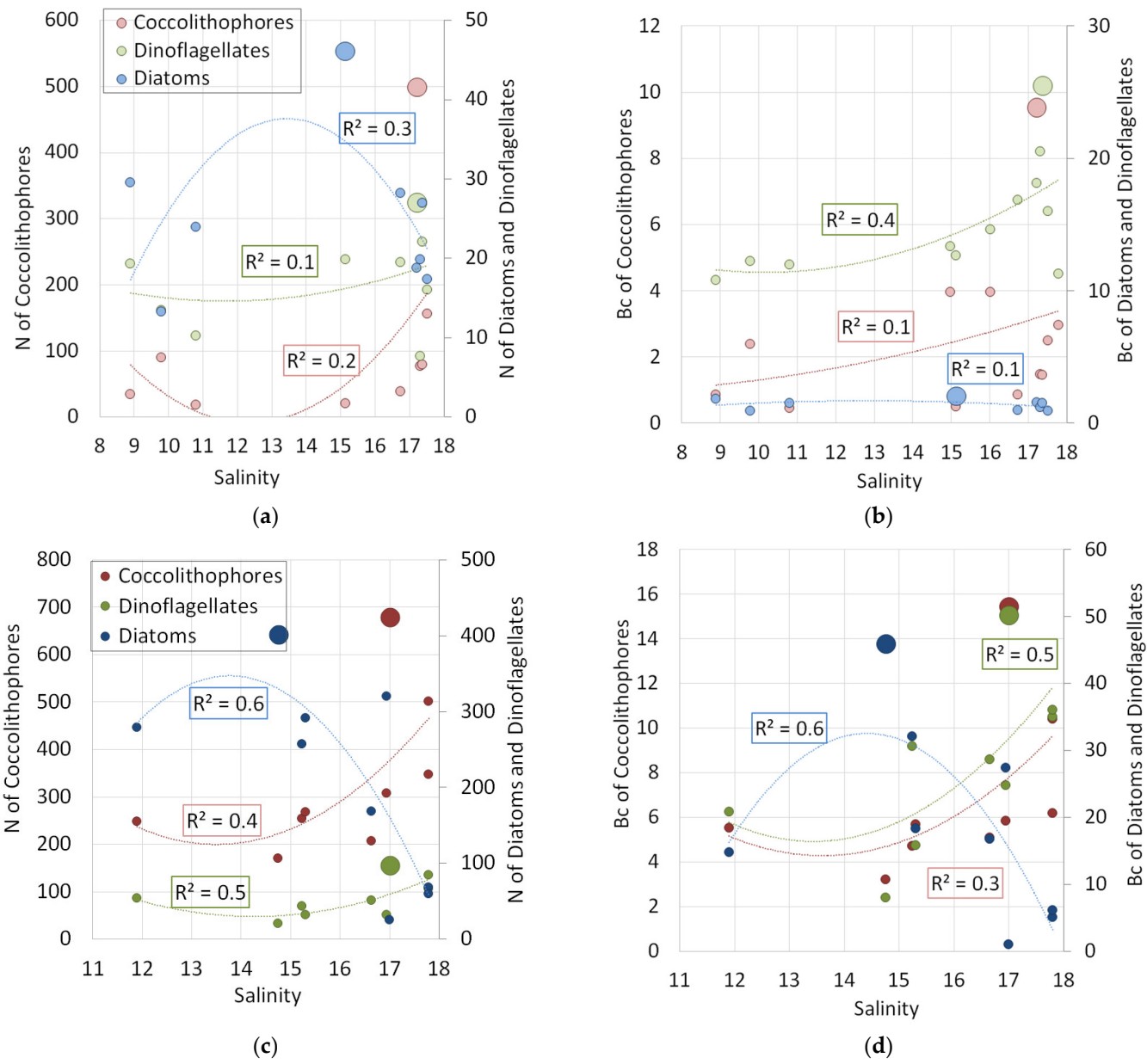

**Figure 6.** Variability of abundance (N, $\times 10^3$ cells $L^{-1}$) and carbon biomass (Bc, µgC $L^{-1}$) of different phytoplankton groups in the salinity gradient under river–seawater mixing in different season (**a**); (**b**)—In April (influence of Kodor River), pale circles; (**c**,**d**)—In May–June (influence of the Ashe, Psezuapse, and Shahe rivers), dark circles. Maximum values are marked with large circles.

For both seasons in April and in May–June, the maximum abundance and biomass of diatoms, dinoflagellates, and coccolithophores coincided with a specific salinity range: for diatoms, 14.8–15.2; for dinoflagellates and coccolithophores, 17–17.2.

### 3.2.3. Active Phototrophs in Phytoplankton Community

This trend was even more pronounced in the distribution of biomass of actively photosynthetic cells (Figure 7).

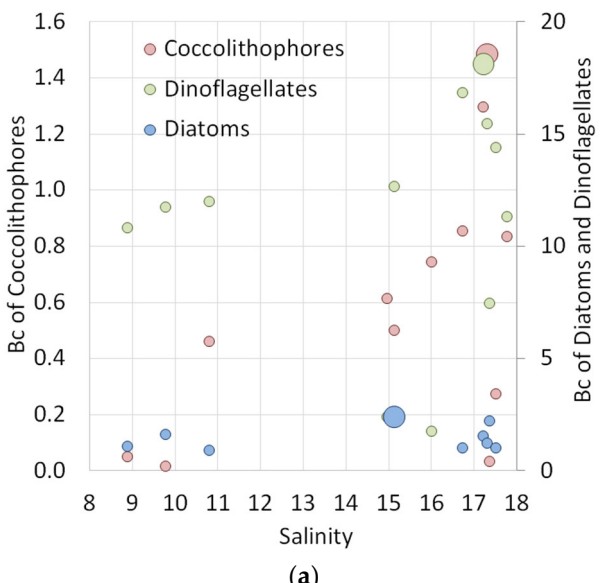

(**a**)

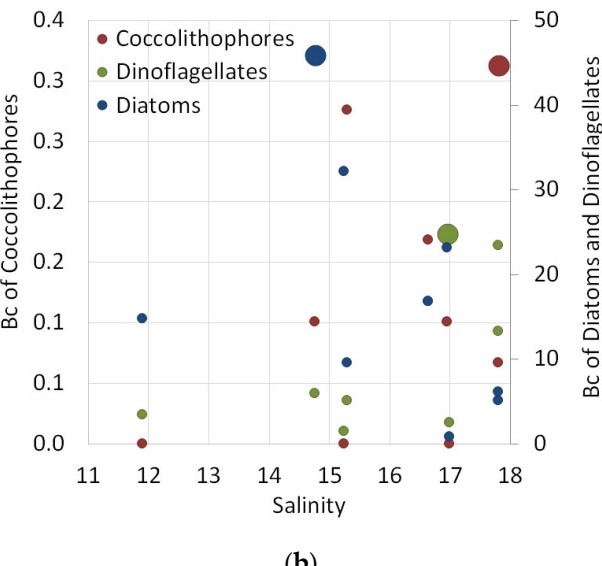

(**b**)

**Figure 7.** Variability of carbon biomass (Bc, µgC L$^{-1}$) of active photosynthetic cells of different phytoplankton groups in the salinity gradient under river–seawater mixing in different seasons: (**a**)—In April (influence of Kodor River), pale circles; (**b**)—In May–June (influence of Ashe, Psezuapse, Shahe rivers), dark circles. Maximum values are marked with large circles.

Among active photosynthetic cells, the dominant species with maximum of abundance and biomass in April (in the Kodor influenced area): among diatoms—*Skeletonema costatum*, *Chaetoceros wighamii*, *C. abnormis* (salinity 15.1), among dinoflagelates—*Alexandrium tamarense*, *Tripos muelleri*, *Gymnodinium* sp. (salinity 17.2), and from coccolithophores—*Emiliania huxleyi* (salinity 17.3). In May–June (in the areas under influence of *the* Ashe, the Psezuapse, and the Shahe)—among diatoms—*Chaetoceros compressus*, *C. curvisetus*, *C. neogracilis*, *Leptocylindrus danicus* (salinity 14.8), among the dinoflagellates are *Prorocentrum cordatum*, *P. micans*, *Scrippsiella acuminata*, *Gonyaulax digitale* (salinity 17), and from coccolithophores—*Emiliania huxleyi* (salinity 17.7).

### 3.2.4. Distribution Patterns of Different Groups

When considering the spatial distribution of carbon biomass of different groups of planktonic microalgae, chl *a*, *PP*, and nutrients (nitrates, ammonium, and phosphates) along the salinity gradient, the zone of increasing biomass of diatoms was called the Diatom Production Zone, and the zone of increasing biomass of dinoflagellates and coccolithophores was named as Dinoflagellate/Coccolithophore Production Zone (Figure 8). The Diatom Production Zone was located within the salinity range of 14.8–15.2. The increase in biomass of diatoms was accompanied by a significant decrease in the nutrient content. At the same time, the chl *a* and *PP* values were generally similar to the adjacent regions. In May–June, with a more pronounced maximum of the diatoms biomass, this area corresponded to a higher level of *PP* compared with the April situation. The zone of increased biomass of dinoflagellate/coccolithophores corresponded to a salinity 17–17.7. The highest *PP* rates and chl *a* concentration were observed here in both seasons: up to 28.5 µgC L$^{-1}$ day$^{-1}$ and

2.4 µg L$^{-1}$ in April, and up to 31.4 µgC L$^{-1}$ day$^{-1}$ and 1.1 µg L$^{-1}$ in May–June. In April, an increase in the production characteristics in the Dinoflagellate/Coccolithophore Production Zone was not accompanied by significant changes in the nutrients concentrations; in May–June, this area corresponded to increased concentrations of all nutrients compared with April.

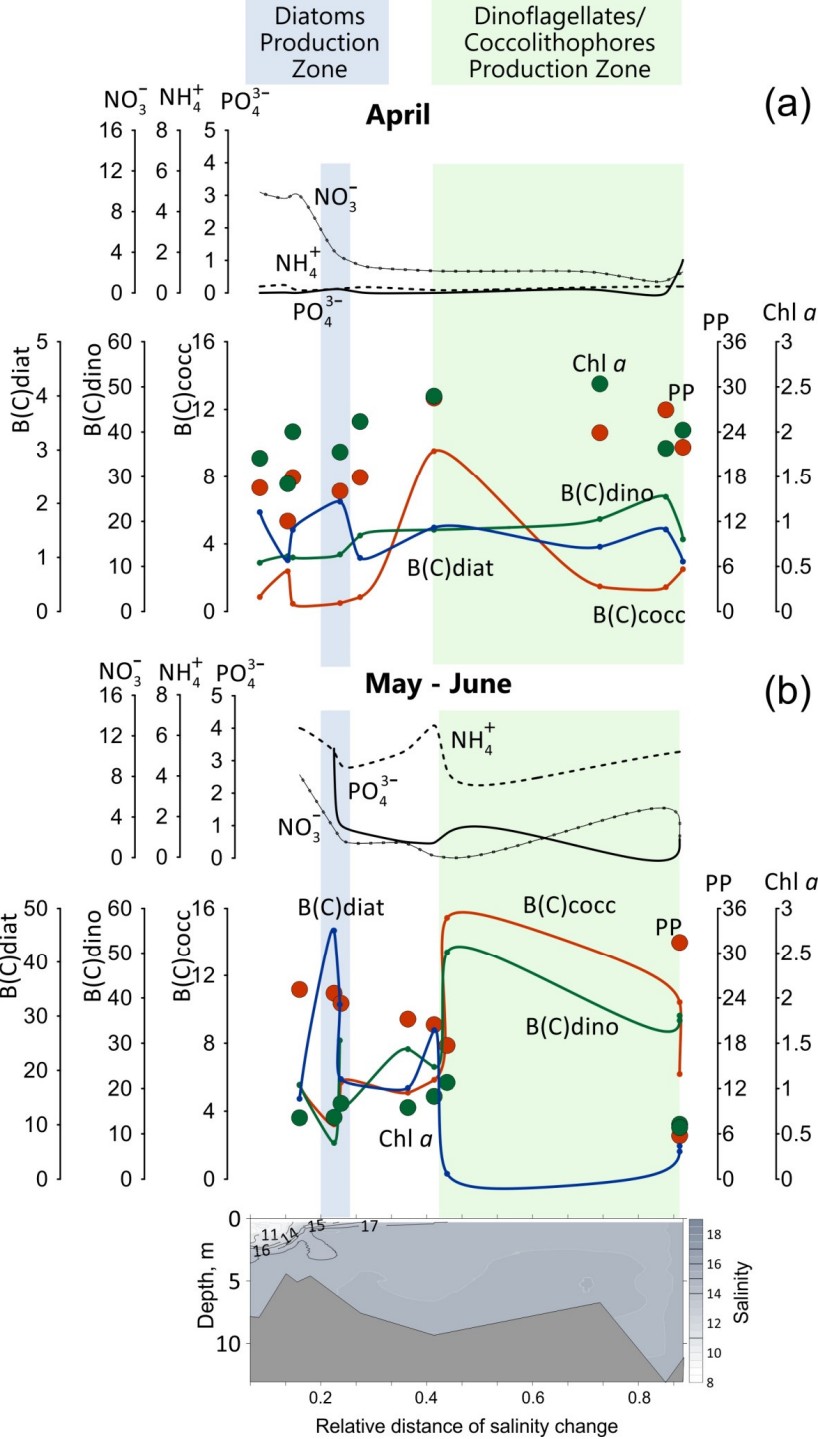

**Figure 8.** Scheme of spatial distribution of Carbon Biomass of Diatoms (B(C)diat, µgC/L), Dinoflagellates (B(C)dino, µgC L$^{-1}$), and Coccolithophores (B(C)cocc, µgC/l), chl *a* (µg L$^{-1}$) and the primary production (*PP*, µgC L$^{-1}$ day$^{-1}$) as well as nutrients—nitrogen in the forms of NO$_3^-$ (µM), NH$_4^+$ (µM), and phosphorus in PO$_4^{3-}$ (µM) in the surface layer along the salinity gradient formed under the influence of small rivers: (**a**)—in April and (**b**)—May–June.

Owing to the coincidence of the maximum abundance and biomass of diatoms, dinoflagellates, and coccolithophores with a narrow salinity range, five areas under the influence of oligotrophic rivers are distinguished in Table 3. Based on surface salinity, the following zones were determined: "Near the river mouth" with a salinity less than 12, "Diatom production" with a salinity range of 14.8–15.1, "Transition" with a salinity of 15.5–17, "Dinoflagellate/Coccolithophore production" with a salinity of 17–17.5, and "Marine zone" with a salinity greater than 17.7. We have observed the decrease of quantitative characteristics of diatoms, dinoflagellates, and coccolithophores both in nitrate-depleted "Transition" and "Marine zone" and in the nutrient-enriched "Near the Mouth River" zone (Figure 8). This effect was more pronounced in May–June compared to April.

**Table 3.** Biomass (Bc, Carbon Biomass; Bw, Wet Weight Biomass, is given in brackets) of leading phytoplankton groups, *PP*, chl *a*, Fv/Fm, and hydrophysical and hydrochemical characteristics in described zones under the influence of oligotrophic rivers in April and May–June. NMR, Near mouth of river zone; DP, Diatom production zone; TR, Transition zone; DCP, Dinoflagellate/Coccolithophore production zone; M, Marine zone.

| | | | NMR | DP | TR | DCP | M |
|---|---|---|---|---|---|---|---|
| **April (Kodor Influence)** | Bc, µgC L$^{-1}$ (Bw, µg L$^{1}$) | Stations | 3a, 6a, 7a | 4a | 2a | 5a, 9a, 11a | 13a |
| | | Salinity | <12 | 14.8–15.1 | 15.5–17 | 17–17.5 | >17.7 |
| | | Temperature, °C | 9.5–9.6 | 10 | 10.2 | 9.9–10.2 | 10 |
| | | Diat | 0.9–1.8 (14.4–30.8) | 2 (24.1) | 1 (14.6) | 1.2–1.6 (22.5–25.5) | 0.9 (12.1) |
| | | Dino | 10.8–12.3 (70.3–106.1) | 12.7 (86.4) | 16.8 (139) | 18.1–25.5 (144.1–424) | 16.1 (144.2) |
| | | Cocc | 0.9–2.4 (2–15.4) | 0.5 (3.2) | 0.8 (5.5) | 1.4–9.5 (9.1–62) | 2.5 (15.7) |
| | Nutrients, µM | N-NO$_3$ | 9.4–9.9 | 3.6 | 2.7 | 1.2–2.2 | 2.1 |
| | | N-NH$_4$ | 0.2–0.4 | 0.2 | 0.3 | 0.1–0.3 | 0.3 |
| | | P-PO$_4$ | 0–0.1 | 0.1 | 0 | 0–0.1 | 1 |
| | Function | chl *a*, µg L$^{-1}$ | 1.4–2 | 1.8 | 2.1 | 1.8–2.4 | 2 |
| | | *PP*, µgC L$^{-1}$ day$^{-1}$ | 12–17.9 | 16.1 | 17.9 | 23.9–28.5 | 21.9 |
| | | Fv/Fm | 0.55–0.59 | 0.53 | 0.59 | 0.62–0.65 | 0.58 |
| **May–June (Ashe, Pesetuapse, Shahe Influence)** | Bc, µgC L$^{-1}$ (Bw, µg L$^{1}$) | Stations | 24 | 3,25,26 | 16,27 | 28 | 15,30 |
| | | Salinity | <12 | 14.8–15.1 | 15.5–17 | 17–17.5 | >17.7 |
| | | Temperature, °C | 21.8 | 22.6–23.1 | 23–23.2 | 22.8 | 22.5–22.8 |
| | | Diat | 14.8 (186.3) | 32.1–45.9 (338.6–694.2) | 16.8–27.4 (250.1–442.5) | 1 (12) | 5.1–6.2 (76.1–96.4) |
| | | Dino | 20.8 (128.7) | 8.0–30.6 (55.6–226.8) | 15.8–24.8 (168.8–201.9) | 50.2 (367.8) | 35–36.1 (209.1–243.9) |
| | | Cocc | 5.5 (34.9) | 3.2–4.7 (20–35.8) | 5.1–5.8 (32.7–36.3) | 15.4 (100.4) | 6.2–10.4 (38.1–65.4) |
| | Nutrients, µM | N-NO$_3$ | 8.2 | 2.3 | 0.2–1.8 | 0 | 1–1.6 |
| | | N-NH$_4$ | 6.4 | 5.2 | 4.5–6.5 | 4.3 | 5.2 |
| | | P-PO$_4$ | | 3.4 | 0.5–1 | 0.8 | 0.4–0.6 |
| | Function | chl *a*, µg L$^{-1}$ | 0.7 | 0.7 | 0.8–0.9 | 1.1 | 0.6 |
| | | *PP*, µgC L$^{-1}$ day$^{-1}$ | 25.1 | 24.6 | 20.5–23.3 | 17.7 | 5.8–31.4 |
| | | Fv/Fm | 0.56 | 0.48–0.52 | 0.5–0.66 | 0.58 | 0.57 |

### 3.2.5. Features of Vertical Distribution of Phytoplankton

In May–June, additional sampling in the Shahe River plume from the bottom layer allowed us to establish that coccolithophore cells were concentrated at the bottom at depths of only 3–8 m (Figure 9a). Also, diatoms developed on the surface. The dinoflagellates were distributed relatively uniformly. The photosynthetic activity of algae at the bottom decreased significantly (Figure 9b): PP decreased by about eight times; chl *a* by about two

times. The maximum quantum efficiency of PSII (Fv/Fm) was less than 0.4. A pronounced decrease in the production characteristics of phytoplankton and a low level of Fv/Fm indicated the processes of pheophytinization, cell death, and detrital formation [59].

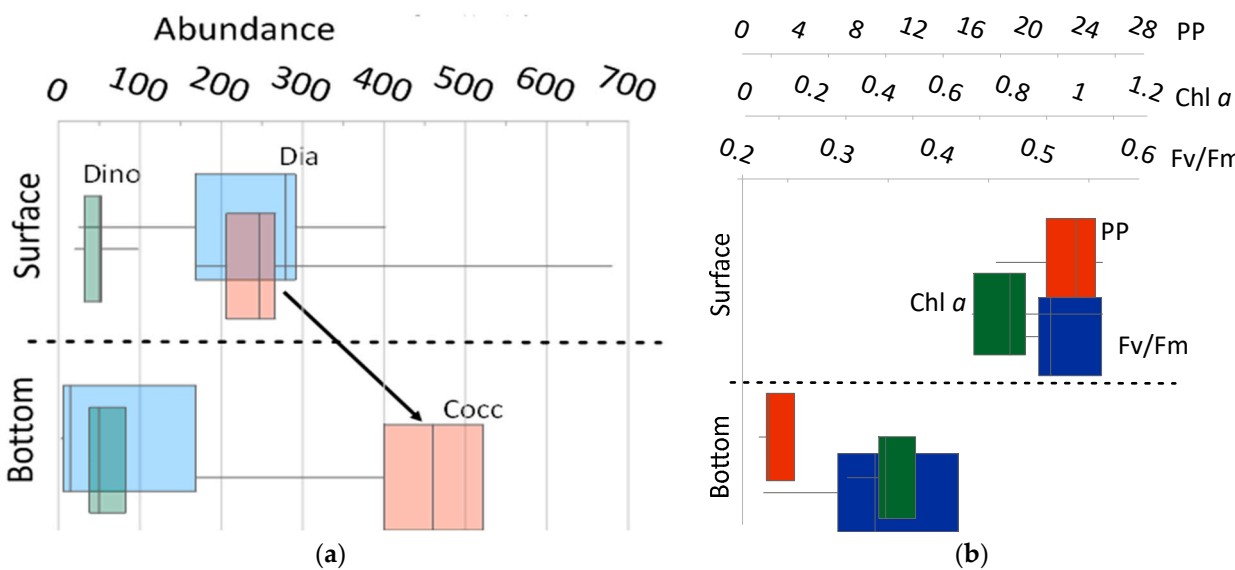

(**a**)  (**b**)

**Figure 9.** Variability range of phytoplankton abundance and characteristics of functioning in surface and bottom layers in the zone of influence of the Shahe River (May-June): (**a**) abundance of different algae groups ($\times 10^3$ cells L$^{-1}$), and (**b**) primary production (*PP*, µg L$^{-1}$ day$^{-1}$), chl a (chl a, µg L$^{-1}$), the maximum quantum efficiency of PSII (Fv/Fm). The bottom and top of the box show the 25th and 75th percentiles, respectively. The vertical line into the box shows the median, whiskers extend to the minimum and maximum values. The boxes are offset from each other for better visualization.

### 3.2.6. Distribution of DOC and n-Alkanes

The DOC concentration in the zone of influence of the Kodor River was 2.9–5.8 mg L$^{-1}$ (or 244.2–481.7 µM). The maximum DOC concentration was found at a salinity of 15.1 (Figure 10a) associated with the zone of increased diatom biomass.

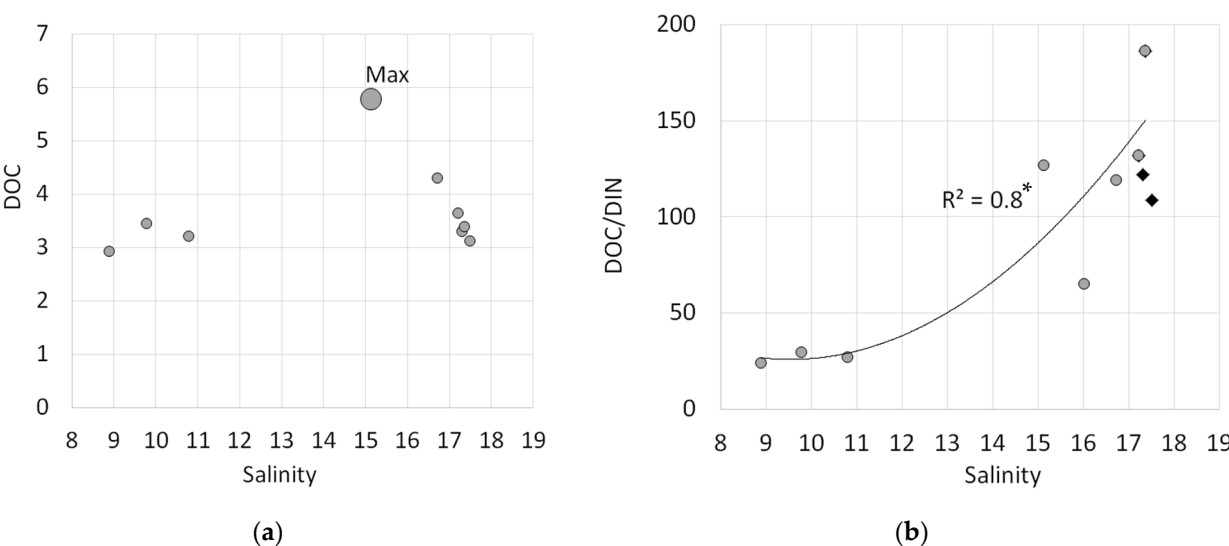

(**a**)  (**b**)

**Figure 10.** DOC distribution in the zone of influence of Kodor River (April): (**a**) DOC distribution (mgC L$^{-1}$) in the salinity gradient and (**b**) ratio of DOC to DIN (nitrogen in the forms of NO$_3$ + NO$_2$ + NH$_4$) plotted in the salinity field. * Regression line for gray circles only. Black rhombi show decrease in ratio in the marine area.

In the river–seawater mixing zone, the ratio of DOC to DIN (nitrogen in the forms of $NO_3^- + NO_2^- + NH_4^+$) showed an increase with decreasing inorganic nitrogen concentration in the salinity range of 15–17.3 (Figure 10b). Farther from the outer boundary of the plume, with a salinity greater than 17.3, this ratio begins to decrease.

The normalized DOC for the total phytoplankton carbon biomass increased with an increase in the diatom carbon biomass and decreased with an increase in the dinoflagellate and coccolithophore carbon biomass (Figure 11).

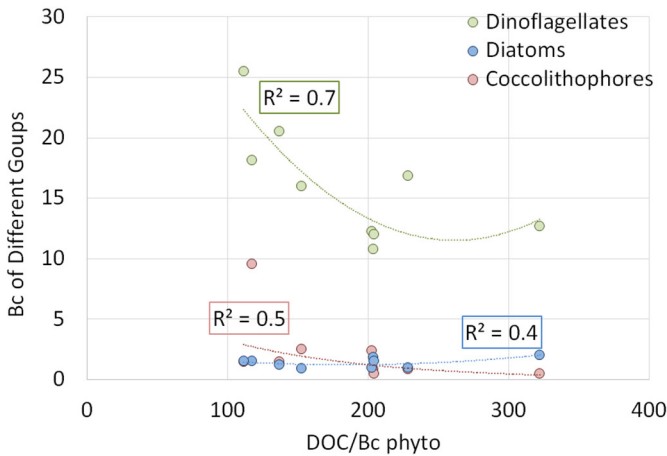

**Figure 11.** Relationship between biomass of different phytoplankton groups and DOC normalized for total phytoplankton biomass.

The surface POM content varies from 0.15 mg $L^{-1}$ in the transition zone to 0.58 mg $L^{-1}$ in the Kodor River. n-Alkane concentrations ($C_{13}$-$C_{31}$) are the same as the lipid content and display a wide range from 0.41 μg $L^{-1}$ in transition zone to 5.69 μg $L^{-1}$ in the Kodor River. The molecular distribution of the n-alkane fraction of all the samples from different zones shows a strong monomodal distribution of homologs (Figure 12) with an increased proportion of low molecular weight homologs ($\Sigma C_{13-24}$ = 93%, on average). The predominant marine hydrocarbons n-$C_{15}$ and n-$C_{17}$ accounted for 25 and 12.5% on average, respectively, of the total n-alkanes. Significant changes in the n-alkane distribution were found the sampling station in the transition zone located between the identified zones of increased biomass of diatoms and dinophytes/coccolithophorids (Figure 12). In contrast to other stations, the major n-alkanes here are n-$C_{17}$ of marine origin (15.7%) and n-$C_{18}$ (15.1%) and n-$C_{20}$ (13.3%) of bacterial origin. The odd-to-even predominance index of n-alkanes in the transition zone is close to 1 (OEP = 0.97), while at sampling stations from other areas, it is lower, reaching 0.75, on average.

### 3.3. Influence of the Mesotrophic Tuapse River

Under the influence of the mesotrophic Tuapse River in late May to early June in the coastal zone, we investigated a small salinity range from 16.2 to 18.1. The phytoplankton quantitative characteristics in the surface layer were $719.9 \times 10^3$–$2687.5 \times 10^3$ cells $L^{-1}$ and 63.2–82.3 μgC $L^{-1}$ (or in a wet unit weight 558.2–1008.8 mg $L^{-1}$). The largest values corresponded to the area of less salinity 16.2. Diatoms were the dominants here with the following photosynthetically active state species: *Thalassiosira* spp., *Leptocylindrus minimus*, *Chaetorceros wighamii*, and *Pseudo-nitzschia pseudodelicatissima*. These species of *Thalassiosira* were found only in the zone of influence of the Tuapse River. In this area, the highest chl *a* and *PP* values were recorded, i.e., 9.1 μg $L^{-1}$ and 330.4 μgC $L^{-1}$ day$^{-1}$, respectively. At a salinity of >17.7, dinoflagellates and coccolithophores dominated, and the chl *a* and *PP* values decreased significantly, i.e., less than 1.1 μg $L^{-1}$ and 16.8 μgC $L^{-1}$ day$^{-1}$. It was dominated by a mixed community of dinoflagellates *Cochlodinium* sp., *Prorocentrum* spp., *Gymnodinium* sp., *Tripos muelleri*, *Scrippsiella acuminata*, young peridineae, and

coccolithophore *Emiliania huxleyi*. *Prorocentrum cordatum*, *Gymnodinium* sp., and *Emiliania huxleyi*, which were in the active state.

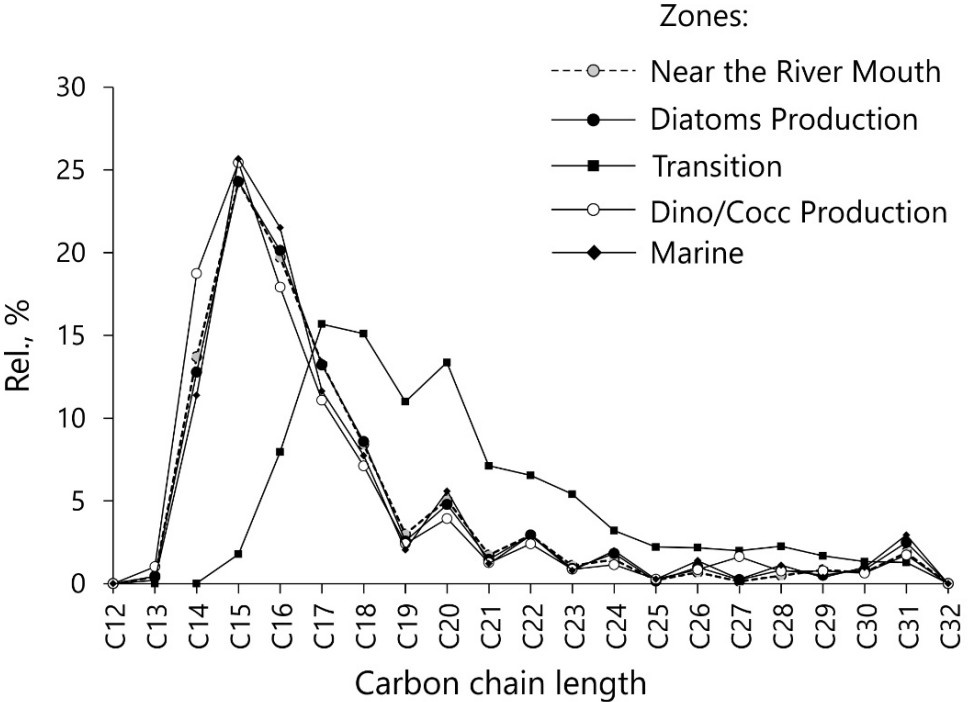

**Figure 12.** Relative distribution of n-alkanes in the surface layer in different zones under influence of the Kodor River in April.

## 4. Discussion

Studies of biological processes in shelf and, in particular, coastal marine zones are very difficult due to the significant spatial variability of the environmental factors. Precision sampling with accurate selection of surface salinity allowed us to obtain the characteristics of the structural and functional organization of phytoplankton in April and May–June in the zones of influence of Caucasian rivers in the NE Black Sea.

In the zones of interaction of oligotrophic rivers with the marine zone, the species richness (90 species and higher rank taxonomic units for April and 78 units for May–June) was comparable with the data of long-term studies of coastal areas, (109 taxonomic units [42]) and lower for the entire NE Black Sea (100–160 species [28]). As in other studies [28,29,36,42], the largest number of species in phytoplankton belonged to diatoms and dinoflagellates. In our studies, in April, the shares of these groups in the total species richness were approximately equal (at a level of 40%). In May–June, the share of dinoflagellate species increased to 53%, while the share of diatoms decreased to 33%. Previously, the predominance of mainly dinoflagellates in the species richness was noted at the level of 60–75% in spring and summer [29,36].

In the zones of influence of oligotrophic rivers, the dominant photosynthetically active species were as follows: In April, *Skeletonema costatum*, *Chaetoceros wighamii*, *C. abnormis*, *Alexandrium tamarense*, *Tripos muelleri*, *Gymnodinium* sp., *Emiliania huxleyi*; In May–June, *Chaetoceros compressus*, *C. curvisetus*, *C. neogracilis*, *Leptocylindrus danicus*, *Prorocentrum cordatum*, *P. micans*, *Scrippsiella acuminata*, *Gonyaulax digitale*, *Emiliania huxleyi*. In the zone of influence of the mesotrophic Tuapse River, *Thalassiosira* spp., *Leptocylindrus minimus*, *Chaetoceros wighamii*, *Pseudo-nitzschia pseudodelicatissima*, *Tripos muelleri*, *Cochlodinium* sp., *Prorocentrum cordatum*, *P. micans*, *Scrippsiella acuminata*, *Gymnodinium* sp., young peridineae, and *Emiliania huxleyi* prevailed. All these species are typical of the Black Sea coastal flora and have been noted in previous studies of phytoplankton in the NE Black Sea [29,36,37,40,42]. The presence of the dinoflagellate *Alexandrium tamarense* and diatom *Pseudo-nitzschia pseu-*

*dodelicatissima* in photosynthetically active states is noteworthy because of their possibilities of producing toxins, ASP (amnesic shellfish poisoning) and PSP (paralitic shellfish poisoning) [16,37]. The maximum abundance of *Alexandrium tamarense* was recorded in April of $1.2 \times 10^3$ cells $L^{-1}$ and of *Pseudo-nitzschia pseudodelicatissima* in early June of $114.2 \times 10^3$ cells $L^{-1}$. This is about four times lower but comparable to the previously obtained estimates of the maximum abundance of these species in the coastal waters of the NE Black Sea: for *Alexandrium tamarense*, $4.42 \times 10^3$ cells $L^{-1}$ in February, and for *Pseudo–nitzschia pseudodelicatissima*, $461 \times 10^3$ cells $L^{-1}$ in June [37]. In addition, in the zones of influence of all rivers, the main dominant determining the total phytoplankton abundance was the coccolithophore *Emiliania huxleyi*. Since the end of the last century, this species began to control the quantitative characteristics of phytoplankton in different seasons in the central and shelf parts of the Black Sea [30,32]. The maximum abundance of this species, 6 million cells $L^{-1}$, was recorded in June on the shelf near Sochi [29]. In our studies, the maximum abundance of *Emiliania huxleyi* was $498.8 \times 10^3$ cells $L^{-1}$ in April and $679.9 \times 10^3$ cells $L^{-1}$ in May–June.

The average value of the recorded total phytoplankton abundance in the zones of influence of small rivers in April was $207.78 \times 10^3$ cells $L^{-1}$ (SD $142.37 \times 10^3$ cells $L^{-1}$) and corresponded to the average value of coastal phytoplankton communities in April ($214.1 \times 10^3$ cells $L^{-1}$) [36]. In May–June, the average value of the total population recorded by us in the zones of influence of oligotrophic rivers was $654.7 \times 10^3$ cells $L^{-1}$ (SD $147.97 \times 10^3$ cells $L^{-1}$), which exceeded more than three times the average for these months, i.e., $119 \times 10^3$ cells $L^{-1}$–$215.7 \times 10^3$ cells $L^{-1}$ [36]. The variability of wet biomass in the surface layer under the influence of oligotrophic and mesotrophic rivers in May–June 2018 was 350.6–1008.8 µg $L^{-1}$ (averaging 577.5 µg $L^{-1}$, SD 211.8 µg $L^{-1}$). This range fit that of the long-term (2002–2012) variability for May–June of wet biomass for the upper layer in the coastal zone with the dominance of coccolithophores, 150–1400 µg $L^{-1}$ [32], and were significantly higher than the range of 109.1–369.2 µg $L^{-1}$ for the median shelf of 50–100 m [29]. In May–June 2018, the ranges of chl *a* in the surface layer in the zones of influence of small rivers, 0.6–9.1 µg $L^{-1}$ (average 1.6 µg $L^{-1}$, SD 2.6 µg $L^{-1}$), and *PP*, 5.8–330.4 µgC $L^{-1}$ day$^{-1}$ (mean 51.5 µgC $L^{-1}$ day$^{-1}$, SD 98.2 µgC $L^{-1}$ day$^{-1}$), exceeded the ranges of long-term variability in May–June for the coastal zone of the NE Black Sea. For this season, the long-term chl *a* values varied within 0.1–0.9 µg $L^{-1}$ (mean value of $0.4 \pm 0.03$ µg $L^{-1}$), and *PP* changed within 2.7–34.9 µgC $L^{-1}$ day$^{-1}$ (mean value of $14.7 \pm 1.4$ µgC $L^{-1}$ day$^{-1}$) [32]. The average long-term values were three to four times lower than those we obtained for coastal marine zones influenced by small rivers in the NE Black Sea.

On a very short spatial scale (from 500 m to 4.5 km), the variability of the quantitative and functional characteristics of phytoplankton in the zones of influence of oligotrophic rivers were significant: seven times for abundance, two times for carbon biomass (wet biomass four times), almost two times for chl *a*, 2.4 times for *PP* in April; in May–June, two times for abundance, 1.5 times for carbon biomass (two times for wet biomass), two times for chl *a*, six times for *PP*. The variability in the zone of influence of the mesotrophic Tuapse River was almost four times for abundance, 1.5 times for biomass (two times for wet biomass), 15 times for chl *a*, and 43 times for *PP*.

At the nearshore zone near the mouth of the river and along the entire salinity gradient in the mixing zone, the phytoplankton community was formed of marine species. The share of freshwater phytoplankton decreased sharply from the mouth to a salinity of 9–11, and it was in an inactive state, and there were pheophytinized cells or empty frustules. The absence of photosynthetically active freshwater species in the coastal zone of the Black Sea near river mouths can be explained by the formation of a sharp salinity gradient here (from 0 to 11). Such high salinity gradients occur due to extremely high flow velocities (2.4–2.6 m s$^{-1}$) in the upper layer of the sea near the mouths of small Caucasian rivers [60]. The disappearance of freshwater species confirms the importance of the salinity boundary at 4–7 as a critical point of physiological stress for freshwater species [61]. Seaward with

smoother spatial gradients in salinity from 15 to 17 and a decrease in current velocities at the plume periphery of about three times (0.6–0.8 m s$^{-1}$) [60], and the penetration of seawater into the plume zone [62] can bring marine phytoplankton into it and stimulate the development of diatoms. During each survey in April and May–June, on a small spatial scale (less than 1 km), we revealed an increase in photosynthetically active cells of diatoms as the dominant group at a salinity of about 15. Marine diatoms developed near the outer boundary of the plume under a constant influx of fresh nitrates and ammonium from river runoff [24]. The maximum concentration of DOC was at a salinity of 15 with a two-fold increase in concentration compared to the background values, from 2.9 to 5.8 mg L$^{-1}$ (or from 244.2 to 481.7 μM). Previous studies show a value of 239 μM as the background for the marine zone of the Black Sea [23]. For a salinity 15–17, the DOC concentration increases with decreasing of nitrogen content (Figure 8). Probably, the increase in the DOC content in the surface is associated with planktonic diatoms activity. Further, at a salinity of 17–17.7, changes in dominants from diatoms to dinoflagellates and coccolithophores were observed. Above a salinity of 17, the ratio of DOC to nitrogen decrease and most likely the produced DOC can be consumed by other groups of phytoplankton like coccolithophores, as shown in [63,64]. Thus, in the marine zone near the plume boundary at a salinity of 17, the development of coccolithophores can be controlled not only by a change in the N/P ratio, as shown for the seasonal development of dominant diatoms or coccolithophorids [29,31,32], but also by enrichment of this zone in DOC. This seems all the more likely due to the n-alkane composition in the surface layer. The abundance and molecular composition of n-alkanes are used as indicator of organic matter genesis and its transformation [65]. The results of the n-alkane study in the surface POM samples from Kodor River runoff in April agree with the biological and hydrochemical features of different zones described above (Figure 8, Table 3). Our data show the presence of "fresh" organic matter, synthesized by phytoplankton, with a low degree of biodegradation. An exception is the transition zone between areas where diatoms and dinoflagellates/coccolithophores prevailed, which shows a different group distribution of n-alkanes. The minimum concentrations of POM and n-alkanes are found here. The distribution of molecular markers shows the presence of more biodegraded organic matter. The input of bacterially derived components is 59%, in contrast to other zones, where this value is 39% on average. We can conclude that the organic matter on transition zone has a distinct genesis from other zones and is more transformed by microbiological processes. In a narrow zone with more biologically transformed organic matter at a salinity of 16.7, there was also an increase in DOC content, i.e., 4.3 mg L$^{-1}$ or 358.3 μM, with respect to the background values.

Such a mechanism of DOC formation by diatoms in the outer area of the plume and DOC consumption in the marine zone adjacent to it can explain the previously obtained correlations of coccolithophore development in the coastal zone with precipitation rates the day before [32]. After several days of heavy rains, an increase in abundance of coccolithophores was observed for several weeks. Due to heavy rains, an increase in small river runoff enriches the nearshore zone with nitrogen and phosphorus and supports the development of diatoms. This in turn increases DOC production at the outer boundary of the plume, which creates favorable conditions for the development of coccolithophores in the zone adjacent to it. The increase in biological activity in the marine zone in direct contact with river plumes is also confirmed by long-term data on the variability of the distribution of the oxygen concentration in the surface layer [22]. Thus, it was shown that at a small distance from the coastline, there was an increase in oxygen compared to river mouth and marine nearshore zone. There were local processes associated with the boundary of this zone located only 1.5–2 km from river mouths.

## 5. Conclusions

Accurate water sampling with precise reference to surface salinity allowed us to establish that small river runoff significantly impacts coastal phytoplankton communities in the NE Black Sea. In the coastal zone, in areas affected by small rivers at a small

spatial scale of 500 m to 4.5 km, the variability of the abundance, biomass, production characteristics, and change of dominant groups is comparable to or exceeds seasonal and interannual variations. The study has demonstrated the role of marine planktonic microalgae in enriching the coastal zone of the NE Black Sea with organic matter under the influence of small rivers and referencing the functioning of different groups of algae to different salinity values. The constant enrichment of the coastal zone with nutrients from small rivers leads to the development, first, of marine diatoms at a salinity of about 15, and then, of dinoflagellates and, in particular, coccolithophorides in the marine zone in contact with the outer boundary of the plume. Obviously, these processes are constant and occur throughout the growth season, possibly even in winter. The accumulation of the most productive calcifying organisms on earth coccolithophores in the bottom, in comparison with the surface layer under river–seawater mixing conditions, may evidence the significant role of small river runoff in regulating the biological carbon pump and controlling the intensity of processes in the carbon cycle. Their subsequent sinking to depth should modify surface alkalinity in the coastal zone and directly affect air–sea $CO_2$ exchange in the NE Black Sea.

**Author Contributions:** Conceptualization, V.M.S.; methodology, V.M.S., S.A.M. and D.M; resources, V.M.S., S.A.M., N.A.S., P.V.K., V.V.K. and D.N.M.; formal analysis, V.M.S., S.A.M. and N.A.S.; investigation, V.M.S., S.A.M., N.A.S., P.V.K. and V.V.K.; writing, V.M.S., S.A.M. and N.A.S. All authors have read and agreed to the published version of the manuscript.

**Funding:** This research was funded by the Ministry of Science and Higher Education of the Russian Federation, topic no. FMWE-2023-0001.

**Institutional Review Board Statement:** Not applicable.

**Data Availability Statement:** The data are contained within the article.

**Acknowledgments:** The authors are deeply grateful to Nikolay Belyaev for help in processing samples, to Alexander Osadchiev and Roman Dbar for help in organizing the field studies, and to Aaron Carpenter for correcting the English translation. The authors are also grateful to the two anonymous reviewers for their thorough work with the manuscript.

**Conflicts of Interest:** The authors declare no conflict of interest.

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
