# Peer review of "Response of the Coastal Phytoplankton Community to the Runoff from Small Rivers in the Northeastern Black Sea"

_diversity, doi:10.3390/d15070857_

Round 1
Reviewer 1 Report
This study nicely analyzed influence of small Caucasian rivers on composition of major phytoplankton groups. Sampling strategy was conducted well with diverse abiotic and biotic parameters measured. It would be valuable if the study also included changes, differences in taxonomic composition inside each phytoplankton group detected (diatoms, dino....) according to different environmental zones and abiotic factors of the studied area.

Author Response
We are very grateful to the Reviewer for thorough reading of the submitted manuscript and helpful comments. We have accepted almost all the comments and corrected the text and figures accordingly. The authors considered this question, but due to the extremely small distances between stations, the composition of phytoplankton at most stations was similar. The most pronounced differences in the zones were among the dominant species, and we made this the focus of our research. The composition of the dominants in each zone is given in the text of the manuscript. When we identified the species, we used numerous identifying book, including in Russian. The names of algae species, as well as their salinity status (marine or freshwater) have been clarified in accordance with World Register of Marine Species (WoRMS, https://www.marinespecies.org/index.php). The corresponding addition has been made to the section “Materials and Methods”.

Reviewer 2 Report
In general, the manuscript should re-written in case of aims, results and conclusions. The units of some analyzed parameters are lacking, e.g. salinity, ora are inappropriate e.g. for nutrients. There are lacking the statistical analyses.
Therefore the manuscript should be re-written and subbmited after revision once more.
In general, the manuscript requires extensive editing. There are many grammar and style errors.
Author Response
The comments of reviewer are very general and we find it difficult to respond to all of the comments. About the lack of salinity units: in accordance with Unesco, 1985 (The International System of Units (SI) in Oceanography. Techn. Pap. Mar. Sci, 45: 124 pp.) salinity measured by CTD probe is dimensionless and doesn’t need any unit because recalculated from conductivity. As well as we have clarified the forms of nitrogen and phosphorus in Materials and methods and have significantly improved the English.
Reviewer 3 Report
The article is devoted to an important aspect of the impact of river runoff on coastal ecosystems in the region of the northeastern coast of the Black Sea. The slopes of the Caucasus Range are replete with stormy river rivers of small length, which are characterized by overflowing with water in spring or autumn. In addition, the coast of the sea, as well as the lower parts of these rivers, remain very attractive for recreation and tourism. Stormy seasonal runoff and recreation periodically carry muddy or polluted waters to coastal waters, thereby leading to a decrease in the quality of the seacoast. The authors of the article devoted their work to studying the influence of the seasonal runoff of mountain rivers on the removal of pollutants. Research methods involve phytoplankton as an indicator of changes introduced by river runoff at river mouths.
The authors chose periods of increased runoff in spring and autumn and studied both seasonal and interannual changes in water chemistry and phytoplankton. The most associated environmental indicators with the development of phytoplankton, the dynamics of its species composition, and the intensity of development of mass species were determined. The indicators of production and species composition were used as a criterion for ongoing changes in the ecosystem of estuaries. The different trophic status of river waters in the rivers flowing into the sea also influenced the development of mass species in coastal ecosystems.
The work was performed at a fairly good level, the methods used in the collection and processing of samples are quite adequate for the goals set, the mentioned samplings are available in the necessary and enough. All cited works are listed in the list of references and vice versa. The illustrations in the required quantity clearly demonstrate the identified patterns. The article is provided with an adequate bibliographic apparatus. The level of conclusions is justified and quite high.
The article may be published in Diversity after minor revision, which are noted as comments in the attached file. In particular, authors can use updated taxonomic names of algae species when they declared it in the MM part:
Gephyrocapsa huxleyi (Lohmann) P.Reinhardt
Scrippsiella trochoidea (F.Stein) A.R.Loeblich III
Gonyaulax digitale (Pouchet) Kofoid
Chaetoceros wighamii Brightwell
The article is devoted to an important aspect of the impact of river runoff on coastal ecosystems in the region of the northeastern coast of the Black Sea. The slopes of the Caucasus Range are replete with stormy river rivers of small length, which are characterized by overflowing with water in spring or autumn. In addition, the coast of the sea, as well as the lower parts of these rivers, remain very attractive for recreation and tourism. Stormy seasonal runoff and recreation periodically carry muddy or polluted waters to coastal waters, thereby leading to a decrease in the quality of the seacoast. The authors of the article devoted their work to studying the influence of the seasonal runoff of mountain rivers on the removal of pollutants. Research methods involve phytoplankton as an indicator of changes introduced by river runoff at river mouths.
The authors chose periods of increased runoff in spring and autumn and studied both seasonal and interannual changes in water chemistry and phytoplankton. The most associated environmental indicators with the development of phytoplankton, the dynamics of its species composition, and the intensity of development of mass species were determined. The indicators of production and species composition were used as a criterion for ongoing changes in the ecosystem of estuaries. The different trophic status of river waters in the rivers flowing into the sea also influenced the development of mass species in coastal ecosystems.
The work was performed at a fairly good level, the methods used in the collection and processing of samples are quite adequate for the goals set, the mentioned samplings are available in the necessary and enough. All cited works are listed in the list of references and vice versa. The illustrations in the required quantity clearly demonstrate the identified patterns. The article is provided with an adequate bibliographic apparatus. The level of conclusions is justified and quite high.
The article may be published in Diversity after minor revision, which are noted as comments in the attached file. In particular, authors can use updated taxonomic names of algae species when they declared it in the MM part:
Gephyrocapsa huxleyi (Lohmann) P.Reinhardt
Scrippsiella trochoidea (F.Stein) A.R.Loeblich III
Gonyaulax digitale (Pouchet) Kofoid
Chaetoceros wighamii Brightwell
More accuracy in the graphs and units required also.

Minor revision of the paper English is need before acceptance.
Author Response
We are very grateful to the Reviewer for thorough reading of the submitted manuscript and helpful comments. We have accepted almost all the comments and corrected the text and figures accordingly. About the lack of salinity units in accordance with Unesco. 1985 (The International System of Units (SI) in Oceanography. Techn. Pap. Mar. Sci, 45: 124 pp.) salinity measured by CTD probe is dimensionless and doesn’t need any unit because recalculated from conductivity. As well as we have clarified which forms of nitrogen and phosphorus we mean in Materials and methods part and corrected throughout the text. We have changed the link Algaebase to World Register of Marine Species in order to leave the name of coccolithophore Emiliania huxleyi. We know that according to molecular phylogeny, this species is very close to the species of the genus Gephyrocapsa. But not all researchers share the opinion that the species should be renamed due to the fact that it was named in the honor famous scientists Thomas Huxley and Cesare Emiliani. Many of them follow the traditions because numerous publications are concerned to Emiliania huxleyi as important species in ecological studies. We have corrected the names of other species according to World Register of Marine Species.

Round 2
Reviewer 2 Report
The manuscript was improved. However, there is still lacking in the Material and methods the information on calculating the relationship between parameters analysed. Pleas add it to this chapter.
Author Response
We are very grateful to the Reviewer for useful comment. We have added relevant information to 2.3. section of “Materials and Methods” (lines 168-173 ).
"Data analysis, verification of the relationship between the abundance and biomass of various algae groups with total quantitative characteristics of phytoplankton, salinity, and DOC as well as calculation of determination coefficients were performed using linear regression and second-order polynomial regression using Microsoft Excel. We also used statistical graphs of Grapher (Golden Software, LLC) to analyze the vertical distribution of algae abundance and the characteristics of functioning in the surface and bottom layers in the zone of influence of the Shahe River."